# Predictors for False-Negative Interferon-Gamma Release Assay Results in Hemodialysis Patients with Latent Tuberculosis Infection

**DOI:** 10.3390/diagnostics13010088

**Published:** 2022-12-28

**Authors:** Heechul Park, Yun-Jeong Kang, Ye Na Kim, Sung-Bae Park, Jaewon Lim, Ji Young Park, Young Ae Kang, Hyejon Lee, Jungho Kim, Sunghyun Kim

**Affiliations:** 1Department of Clinical Laboratory Science, College of Health Sciences, Catholic University of Pusan, Busan 46252, Republic of Korea; 2Clinical Trial Specialist Program for In Vitro Diagnostics, Brain Busan 21 Plus Program, Graduate School, Catholic University of Pusan, Busan 46252, Republic of Korea; 3Department of Laboratory Medicine, Good Samsun Hospital, Busan 47007, Republic of Korea; 4Department of Internal Medicine, Kosin University Gospel Hospital, Busan 49267, Republic of Korea; 5Department of Biomedical Laboratory Science, Masan University, Changwon 51217, Republic of Korea; 6Department of Internal Medicine, Park Clinic, Busan 46512, Republic of Korea; 7Division of Pulmonary and Critical Care Medicine, Department of Internal Medicine, Severance Hospital, Seoul 03772, Republic of Korea; 8Institute of Immunology and Immunological Disease, Yonsei University College of Medicine, Seoul 03772, Republic of Korea; 9Clinical Vaccine Research Section, International Tuberculosis Research Center, Seoul 03772, Republic of Korea

**Keywords:** TB, LTBI, hemodialysis patients, IGRAs, QFT-GIT

## Abstract

The present study aimed to clinically evaluate the effect of T-cell dysfunction in hemodialysis (HD) patients with latent tuberculosis (TB) infection (LTBI) who were false-negatives in the QuantiFERON-TB Gold In-Tube (QFT-GIT) test. Whole blood samples from a total of 20 active TB patients, 83 HD patients, and 52 healthy individuals were collected, and the QFT-GIT test was used for measuring *Mycobacterium tuberculosis* (MTB)-specific interferon gamma (IFN-γ) level. The positive rate of the IFN-γ release assays (IGRAs) in HD patients was lower than the negative rate. The mean value of MTB-specific IFN-γ level, which determines the positive rate of the IGRA test, was highest in active TB, followed by HD patients and healthy individuals. Among HD patients, phytohemagglutinin A (PHA)-stimulated IFN-γ levels of approximately 40% were 10.00 IU/mL or less. However, there was no low level of PHA-stimulated IFN-γ in the healthy individuals. This reveals that T-cell function in HD patients was reduced compared to healthy individuals, which leads to the possibility that QFT-GIT results in HD patients are false-negative. The clinical manifestations of TB in patients on HD are quite non-specific, making timely diagnosis difficult and delaying the initiation of curative treatment, delay being a major determinant of outcome.

## 1. Introduction

Tuberculosis (TB) is one of the top 10 causes of death worldwide and the leading cause of death from a single infectious agent, *Mycobacterium tuberculosis* (MTB) [1]. There were 10 million new cases of TB worldwide in 2018—5.8 million men (58%) and 3.2 million women (32%) [2], and an estimated one million children; 230,000 children died of TB [3]. Globally, the absolute number of TB deaths among human immunodeficiency virus (HIV)-negative people has fallen by the best estimate of 29% since 2000—from 1.8 million in 2000 to 1.3 million in 2017—and by 5% since 2015 [4]. Diagnosis of MTB infection is an important factor in reducing the number of cases of TB in minimally affected countries, and many studies and experiments are still underway [5]. However, the Republic of Korea still has 33,796 TB patients, the highest rate among Organization for Economic Cooperation and Development (OECD) countries in 2018 [4]. Therefore, the fundamental challenge of how to optimize the evaluation of diagnostics for TB infection, for which there is currently no accurate diagnostic assessment, needs to be addressed.

Two main immunodiagnostic methods currently used for the diagnosis of LTBI are the tuberculin skin test (TST) [6] and the interferon-gamma (IFN-γ) release assays (IGRAs) [7]. These tests work on the principle of in vivo and ex vivo cell-mediated immunity, respectively. The TST is an intradermal injection of tuberculin-purified protein derivatives (PPD) to identify the whole MTB Ag-specific immune response, delayed-type hypersensitivity. Nevertheless, it has been reported that the accuracy measures of TST are often confounded by the Bacillus Calmette Guérin (BCG) vaccination and nontuberculous mycobacteria (NTM) infections [8]. IGRA is an ex vivo whole blood test of the T-cell immune response that measures the T-cell release of IFN-γ, an inflammatory cytokine, following stimulation by Ags specific to the MTB complex (MTBC), including MTB, *M. africanum*, *M. bovis*, *M. bovis* BCG, *M. orygis*, *M. microti*, *M. canetti*, *M. caprae*, *M. pinnipedii*, *M. sucricattae*, and *M. mungi* [9]. MTB-specific Ags used in IGRAs, such as early secretory antigen target 6 (ESAT-6) and culture filtrate protein 10 (CFP-10), and TB 7.7 are not present in all *M. bovis* BCG strains or major environmental mycobacteria, NTMs, and are therefore specific for MTB and respond to T-cells that have had MTB bacteria in their blood [10,11]. IGRAs have shown significant potential to reduce false-positive results, particularly among BCG-vaccinated low-risk populations [11]. However, IGRAs have also been known to have limited sensitivity for immunocompromised individuals and young children. Furthermore, IGRAs cannot differentiate between active TB disease and LTBI [12].

End-stage renal disease (ESRD) is known to compromise cellular immune function, and patients on HD have been shown to have T-cell dysfunction, including abnormal IFN-γ levels, considered to have a pivotal role in controlling LTBI. In fact, among ESRD patients, those on HD had a higher prevalence of TB compared to other patient groups. According to Min et al., the probability of being infected with MTB actively during HD or peritoneal dialysis among ESRD patients increased 4.44 ratio for HD patients and 5.17 ratio for peritoneal dialysis patients, respectively [13]. In ESRD patients and HD patients, there is a possibility that T-cell dysfunction related to adaptive immunity may occur, unlike in normal healthy persons. It is therefore likely that results of the IGRA test for diagnosing LTBI and the interpretation of the results are likely to be affected.

The aim of the present study was to evaluate the effectiveness of LTBI diagnosis using IGRA in a total of 20 patients with active TB, 84 patients on HD, and 52 healthy individuals. The IGRA test was performed by QuantiFERON TB Gold In-Tube (QFT-GIT) test (Qiagen, Hilden, Germany). Characteristics of MTB-specific Ags and phytohemagglutinin A (PHA)-stimulated IFN-γ levels were compared between HD patients and healthy individuals.

## 2. Materials and Methods

### 2.1. Study Design and Locations

This study targeted patients receiving HD for approximately one year, from March 2017 to June 2018. A total of 84 HD patients and 52 healthy subjects were enrolled from Kosin University Gospel Hospital and Catholic University of Pusan, Busan, Republic of Korea, and their whole blood samples were collected and used for the present study. All participants provided written informed consent. The Institutional Ethics Committee of Catholic University of Pusan and Kosin University Gospel Hospital approved the study (approval numbers CUPIRB-2017-036 and KUGH 2017-11-042). The active TB patients group was enrolled from November 2010 to March 2012 at Yonsei University Severance Hospital (third referral hospital) in Seoul, Republic of Korea. All participants provided written consent and were approved by the Yonsei University Severance Hospital Institutional Ethics Committee (approval number 4-2010-0527).

HD patients were more than 31 years old with ESRD and were undergoing more than three months of HD treatment. Those with HIV infection, liver cirrhosis of Child–Pugh class C, cancer, or autoimmune disease and those who had received chemotherapy within the last three months and who had any history of active TB treatment were excluded. In addition, eligible subjects consenting to the study were recruited into the active TB group, which consisted of patients with active pulmonary TB. The diagnosis was confirmed by culturing MTB from respiratory specimens. Individuals with HIV infection, end-stage renal disease, or leukemia/lymphoma, and those who had received anti-TB therapy for more than two weeks or immunosuppressive therapy, including anti-cancer chemotherapy for malignant disease, within two months of enrollment were excluded from the study. Results of IGRA, MTB-specific Ag stimulated IFN-γ release, and PHA-stimulated IFN-γ release in the collected whole blood samples were compared between the two groups, hemodialysis patients and healthy non-TB subjects. Plasma was isolated from whole blood after MTB-specific antigen stimulation and stored at −70 °C until use.

### 2.2. Interferon-Gamma Release Assay with QuantiFERON^®^-TB Gold In-Tube (QFT-GIT) Test

An IGRA test was performed on all participants enrolled in this study. IGRAs are whole blood tests that are able to aid in diagnosing MTB infection by detecting IFN-γ in human plasma samples. QFT-GIT assay (Qiagen) was used according to the manufacturer’s instructions on samples from HD patients and from healthy individuals. Whole blood samples were divided into three QFT-GIT collection tubes for nil (negative control), MTB-specific antigen stimulation, and T-cell mitogen stimulation (positive control). The nil control tube contained only the anticoagulant lithium heparin, used for determining the basal level of immune response of peripheral blood cells. The MTB-specific antigen tube contained anticoagulant and MTB-specific antigens, such as ESAT-6, CFP-10, and TB 7.7, used for detecting MTB-specific immune response. The mitogen control tube contained anticoagulant and PHA, used for detecting immune cell function, as a positive control. Antigen stimulation should be performed within 3 h after peripheral blood collection. Antigen stimulation was performed during 24 h at 37 °C for the QFT-GIT (Qiagen) IFN-γ ELISA assay.

### 2.3. Data Interpretation of Interfero-Gamma Release Assay

The test results were interpreted using GFT-GIT ELISA software (version no. 2.43; Melbourne, Australia), and the cut-offs for diagnosis in the manufacturer’s instructions were used. The test result was considered positive if the TB antigen minus Nil IFN-γ level in the sample well, after stimulation with ESAT-6, CFP-10, and TB 7.7 was above 0.35 IU/mL, irrespective of the result for the positive control well. The test was considered negative if the TB antigen minus Nil IFN-γ level was less than 0.35 IU/mL, and the mitogen minus Nil IFN-γ level was above 0.5 IU/mL. The test result was considered indeterminate if the TB antigen minus Nil IFN-γ level was more or less than 0.35 IU/mL and less than 0.5 IU/mL in the mitogen minus Nil IFN-γ level.

### 2.4. Statistical Analysis

Statistical analysis was performed using GraphPad prism, version 5.00 (GraphPad Software, San Diego, CA, USA). The significant difference in the results between the hemodialysis patients and the healthy individuals, which were predicted to be weaker, was calculated. In particular, the value of mitogen control, which is coated with a PHA that stimulates T-cells nonspecifically to secrete IFN-γ, is calculated as the difference between the hemodialysis patients and the healthy individuals, with 95% confidence intervals (CIs), and the values of these groups were compared using the Mann–Whitney U test. A *p*-value less than 0.05 was considered statistically significant.

## 3. Results

### 3.1. Characteristics of Study Participants

The characteristics of the patients involved in this study are shown in Table 1. The mean age of active TB patients was 31 years (range 21–69) and the male to female ratio was 12:8 (60%:40%) and HD patients was 61.1 years (range 31–90), and the male to female ratio was 43:41 (51.2%:48.8%), while the mean age of the healthy individuals was 23.9 years (range 22–35), and the male to female ratio was 23:29 (44.2%:55.8%). The age of the active TB and HD patients was considerably higher than that of the healthy individuals. In addition, only 7 (35%) and 57 (67.9%) of the patients group had received the BCG vaccine, while all the healthy individuals had received the vaccine. Cases reporting having been in contact with TB patients were 5 (25%) and 13 (15.4%) of the patient group, and there were only 6 (7.1%) patients with radiological lesions among the HD patients. The active TB group was positive in AFB stain and PCR results; no additional radiological lesions were identified.

### 3.2. GFT-GIT Results for Active TB, Hemodialysis Patients, and Healthy Individuals

A comparison of the IGRA test between active TB patients, HD patients, and the healthy individuals revealed a significant difference, as shown in Table 2. All of the active TB groups satisfied the study criteria for IGRA, and all 20 (100.0%) were positive for IGRA. Among 84 HD patients, all satisfied the study criteria for IGRA, 34 (40.4%) were positive for IGRA, 49 (58.3%) were negative, and 1 was indeterminate. In healthy individuals, the sensitivity of the test was very high, as 48 of 52 subjects showed negative results. As a result, except for the active TB patients, who were all positive for IGRA, the QFT-GIT positivity rates of HD patients and healthy individuals were 34/84 (40.4%) and 4/52 (7.6%), respectively, thus reflecting that IGRA positivity was significantly higher in HD patients than in healthy individuals. Indeterminate patients were excluded from subsequent studies to differentiate between LTBI and non-LTBI status in HD patients.

### 3.3. Comparison of MTB-Specific Antigen-Stimulated IFN-γ Production between Active TB Patients, Hemodialysis Patients, and Healthy Individuals

The mean values of MTB-specific Ag-stimulated IFN-γ in active TB patients, HD patients, and healthy individuals were 6.95 IU/mL (8.4 ng/mL), 0.77 IU/mL (1.57 ng/mL), and 0.255 IU/mL (0.308 ng/mL), respectively (Figure 1). These results showed that the IFN-γ value was significantly higher in active TB patients and HD patients with LTBI; however, the T-cell mitogen-stimulated IFN-γ value was much higher in healthy individuals. Based on these results, we confirmed that T-cell function in HD patients was reduced compared to healthy individuals (Figure 2). Table 3 shows the proportion of IFN-γ stimulated with TB antigen and IFN-γ stimulated with PHA in the entire group of participants. In this result, it was confirmed that the value of IFN-γ stimulated with MTB-specific Ags in the active TB patients was significantly higher than other groups. In addition, the rate of IFN-γ stimulated with MTB-specific Ags was higher than 0.5 IU/mL (1.2 ng/mL) in 30 cases (88.2%) and lower than 0.5 IU/mL (1.2 ng/mL) in 4 cases (11.8%). On the other hand, in healthy individuals, there were 4 cases (7.6%) and 48 cases (92.3%), respectively.

### 3.4. Comparison of Results from MTB-Specific Ag Stimulated IFN-γ Levels and PHA-Stimulated IFN- γ Levels between Hemodialysis Patients and Healthy Individuals

The correlation coefficient graph of MTB-specific Ag stimulated IFN-γ levels and PHA-stimulated IFN-γ levels is shown in Figure 3. A total of 85% of patients with active tuberculosis had PHA-stimulated IFN-γ levels below 10 (Figure 3A). Of 83 HD patients, 33 (39.8%) had PHA-stimulated IFN-γ levels below 10, while14 (41.2%) of HD patients with LTBI and 19 (38.8%) non-LTBI HD patients, respectively, also had PHA-stimulated levels below 10 (Figure 3B). On the other hand, none of the healthy individuals had PHA-stimulated IFN-γ levels below 10 (Figure 3C).

## 4. Discussion

TB is one of the top 10 causes of death in the world, and the leading cause from a single infectious agent (ranking above HIV/AIDS) [1]. Millions of people continue to fall sick with TB each year [4]. Therefore, there is a need for a method to quickly diagnose those with a high risk of TB. Delayed diagnosis in this situation is an important issue contributing to the high burden and transmission of TB in developing countries.

In a previous study, the performance of QFT-GIT assay was not excellent in patients with impaired immune function, and the results were unclear [14]. Recently, the relationship between TB and the immune status of patients has been revealed, and directions for diagnosis based on the immune status have been suggested [15]. Another study has shown that IGRA testing is useful in identifying LTBI in patients with rheumatic disease, one of the groups at high risk of TB [16]. The population undergoing HD is expanding and is suggested as a high priority group for LTBI treatment, given their increased TB risk [17]. Immunologically, renal disease (RD) is associated with some disorders in both innate and adaptive immune systems in such a way that both immune activation and immune suppression [18] coexist. Immune response to MTB infection is primarily mediated by T-cells [19].

Recent studies have shown that HD patients have a higher infectivity because their immune system is weakened [20]. This study investigated the risk of TB in patients undergoing HD and in healthy individuals and compared them with LTBI status indicated by QFT-GIT assay. The LTBI positivity was significantly higher in the HD patients than in the healthy individuals. It was confirmed that four IGRA-positive in healthy individuals was common, considering the global population trends of LTBI [21]. Table 3 shows that more IFN-γ is secreted in T-cells that have been exposed to TB in HD patients than in healthy individuals. In HD patients whose mitogen-nil values were less than 10 IU/mL (24 ng/mL), the immune responses of the T-cells were lower than those of the healthy individuals. As shown in Figure 3B, in the healthy individuals, no patients with a PHA-stimulated IFN-γ level lower than 10 were observed. However, in the group of HD patients, 14 (41.2%) and 19 (38.7%) patients, respectively, showed a PHA-stimulated IFN-γ level below 10. These results were similar to the levels of PHA-stimulated IFN-γ in the active TB patients group compared to healthy subjects. These results suggest that PHA-stimulated IFN-γ levels are lower than 10 and that T-cell mediated immunity is lowered, resulting in a false-negative [22]. Other studies have reported an alteration of the adaptive immune system for ESRD patients. For example, in the ESRD patient group, the vaccination response to TB or the TST to diagnose LTBI resulted in weak or no response [23]. Compared with the results of this study, the QFT-GIT assay also shows similar results for ESRD patients.

Impaired cellular immunity may explain anergy and high rates of false-negative results of IGRA in HD patients [24]. In fact, many ESRD patients and HD patients are known to have symptoms of a uremia-induced immune system and an acquired immune system deficiency, resulting in various infections and complications [23]. In vitro studies have shown that uremic milieu caused by ESRD reduces T-cell function and population. It has been shown that impaired T-lymphocyte function in ESRD patients could contribute to an immunodeficiency status in HD patients with uremia [25]. Results of this study also confirm that the IFN-γ concentration in plasma samples is decreased during stimulation by T-cell mitogen, possibly due to uremic milieu in ESRD patients. Therefore, further studies on the correlation between serum concentration of urea levels and QFT-GIT assay results are needed. For ESRD patients, it may be possible to use the serum concentration of urea as an auxiliary index in the interpretation of the QFT-GIT assay results.

In this study, among the negative cases in HD patients, there might be positives, that is, there are likely to be false-negatives due to the immune response of T-cells being reduced. Therefore, in patients with T-cell dysfunction such as HD patients, it is necessary to adjust the cut-off value for the early diagnosis of TB. According to recent studies, many studies have also been conducted to readjust the cut-off value of immunological diagnostic methods due to underlying disease conditions and environmental factors in cancer patients [26,27]. Even in the case of TB, there is a possibility that effective prevention and early diagnosis could be made if the cut-off value of QFT-GIT assay in immunosuppressive or immunocompromised patients is adjusted lower than the normal value.

This study has several limitations. The number of samples and the age distribution in each study group do not match. To demonstrate exactly how T-cell dysfunction in immunocompromised HD patients actually affects QFT-GIT testing, the number of samples and the age distribution of each study group must be matched.

## 5. Conclusions

The reliability of QFT-GIT assay in patients with T-cell-mediated immune responsiveness is not high; additional analyses using alternative immune markers or inflammatory markers, and qRT-PCR that simultaneously detects the mRNA expressions of TB-related proinflammatory immune biomarkers may increase the reliability of the test and be more helpful for the rapid screening and detection of groups at high risk for TB, including HD patients.

## Figures and Tables

**Figure 1 diagnostics-13-00088-f001:**
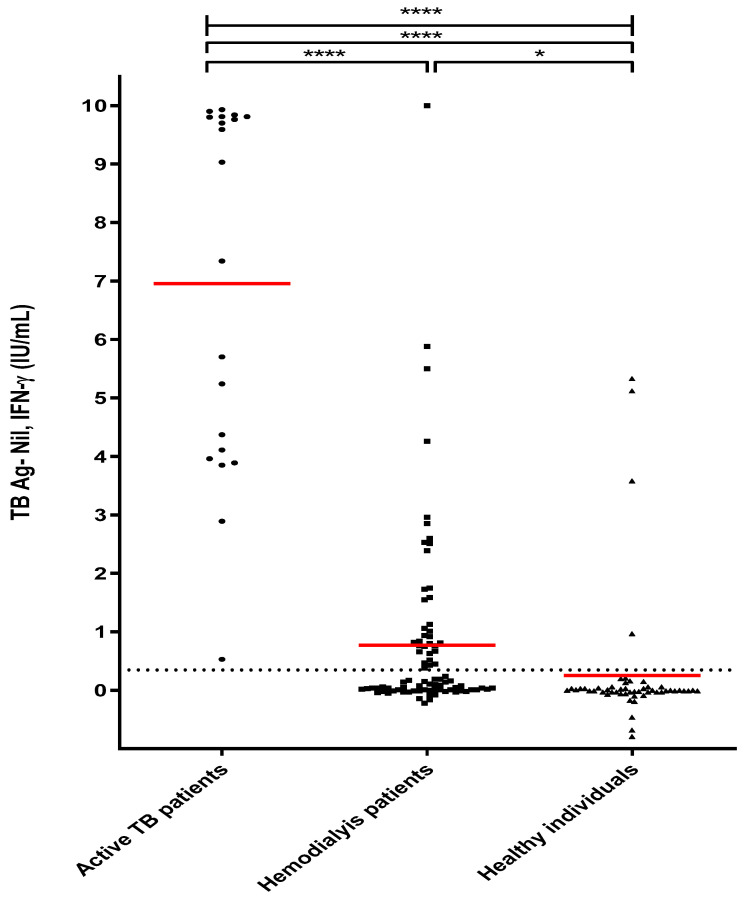
MTB-specific Ag stimulated IFN-γ levels in Active TB patients, hemodialysis patients, and healthy individuals. Note: Whole blood of study subjects (active TB patients group, hemodialysis patients (HD) group, and healthy individuals (HI) group) were stimulated with MTB-specific antigens (early secretory antigen target 6, culture filtrate protein 10, and TB 7.7) for 24 h. The red line in the graph represents the mean value for each group. Active TB patients (*n* = 20); hemodialysis patients (*n* = 83), healthy individuals (*n* = 52); * *p* < 0.05, **** *p* < 0.0001.

**Figure 2 diagnostics-13-00088-f002:**
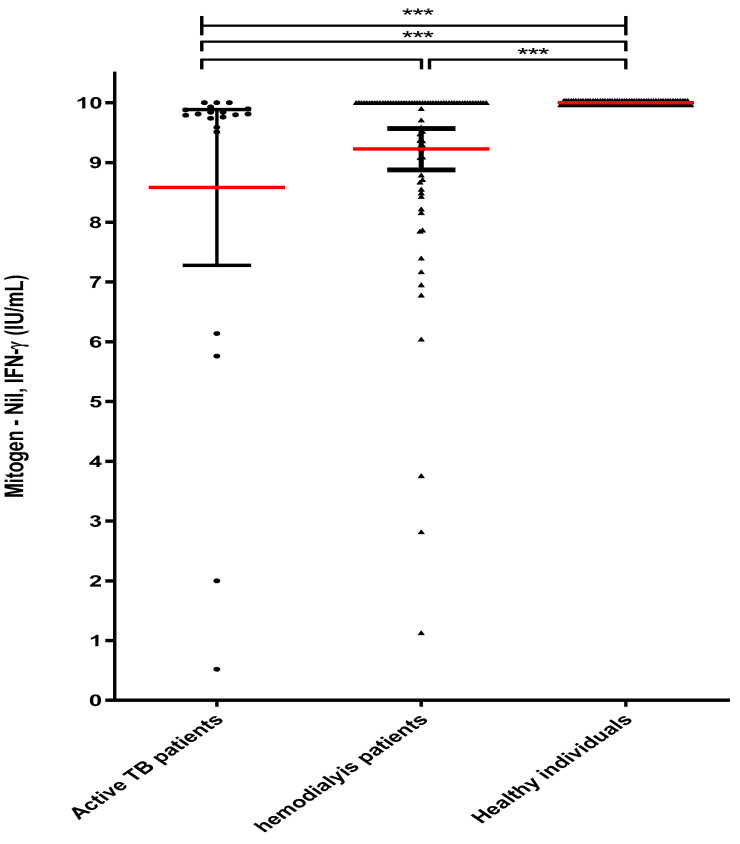
PHA-stimulated IFN-γ levels in hemodialysis patients and healthy individuals. Note: Whole blood of study subjects (active TB patients group, hemodialysis patients (HD) group, and healthy individuals (HI) group) were stimulated with Phytohemagglutinin A for 24 h; The red line in the graph represents the mean value for each group; *** *p* < 0.001.

**Figure 3 diagnostics-13-00088-f003:**
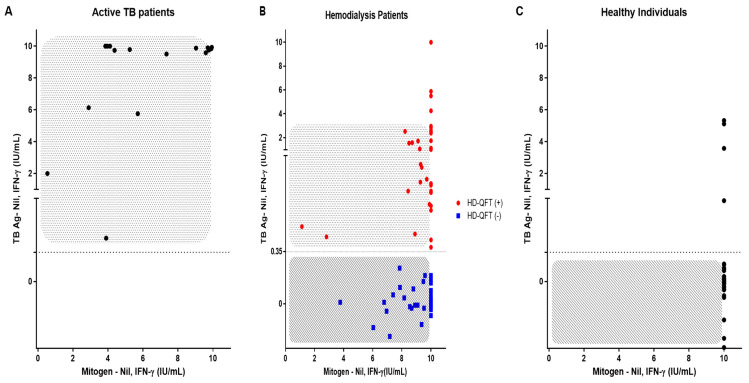
Comparison of results from MTB-specific Ag stimulated IFN-γ levels and PHA-stimulated IFN-γ levels between Active TB patients (**A**) hemodialysis patients (**B**) and healthy individuals (**C**). Note: Correlation data of IFN-γ stimulated with PHA and MTB-specific antigens in hemodialysis patients: red scatter dot is HD with QFT-positive group and blue scatter dot is HD with QFT-negative group; gray box: the part of abnormal T cell capability (85% in active TB patients group, 41.2% in HD with QFT-positive group, 38.8% in HD with QFT-negative group, and no one in HI group).

**Table 1 diagnostics-13-00088-t001:** Characteristics of study participants.

Characteristics		Active TB Patients, *n* (%)	Hemodialysis Patients, *n* (%)	Healthy Individuals, *n* (%)
Age		(median = 31 SD ± 11.1, range = 21–69)	(median = 61.1 SD ± 11.5, range = 31–90)	(median = 23.9 SD ± 2.7, range = 22–35)
	20 years	7 (35.0)	0 (0.0)	50 (96.2)
	30–40 years	11 (55.0)	12 (14.2)	2 (3.8)
	50–60 years	2 (10.0)	52 (62.1)	0 (0.0)
	>70 years	0 (0.0)	20 (23.7)	0 (0.0)
Sex	Male	12 (60.0)	43 (51.2)	23 (44.2)
	Female	8 (40.0)	41 (48.8)	29 (55.8)
Current smoker		-	7 (8.3)	12 (22.2)
BCG scar or vaccination		7 (35.0)	57 (67.9)	52 (100.0)
Contact TB history		5 (25.0)	13 (15.4)	0 (0.0)
Radiological lesions		-	6 (7.1)	0 (0.0)
Total		20 (100.0)	84 (100.0)	52 (100.0)

**Table 2 diagnostics-13-00088-t002:** IGRA results for active TB patients, hemodialysis patients, and healthy individuals.

	IGRA	*n* (%)
Active TB patients (*n* =20)	positive	20 (100.0)
	indeterminate	0 (0.0)
	negative	0 (0.0)
Hemodialysis patients (*n* =84)	positive	34 (40.4%)
	indeterminate	1 (1.1%)
	negative	49 (58.3%)
Healthy Individuals (*n* =52)	positive	4 (7.6%)
	indeterminate	0 (0.0)
	negative	48 (92.3%)

**Table 3 diagnostics-13-00088-t003:** Comparison of IFN-γ production TB antigen stimulated.

	TB Ag-Nil, IFN-γ (IU/mL)	*n* (%)	Mitogen-Nil, IFN-γ (IU/mL)	*n* (%)
Active TB patients (*n* = 20)	≥5	13 (65.0)	<5	2 (10.0)
	>1, <5	6 (30.0)	>5, <10	15 (75.0)
	>0.5, <1	1 (5.0)	≥10	3 (15.0)
	<0.5	0 (0.0)		
Hemodialysis patients QFT-positive (*n* = 34)	≥5	3 (8.8%)	<5	2 (5.9)
	≥1, <5	14 (41.2%)	≥5, <10	12 (35.3)
	≥0.5, <1	13 (38.2%)	≥10	20 (58.8)
	<0.5	4 (11.8%)		
Hemodialysis patients QFT-negative (*n* = 49)	≥5	0 (0.0)	<5	1 (2.0)
	≥1, <5	0 (0.0)	≥5, <10	18 (36.7)
	≥0.5, <1	0 (0.0)	≥10	30 (61.2)
	<0.5	49 (100.0)		
Healthy individuals (*n* = 52)	≥5	2 (3.8)	<5	0 (0.0)
	>1, <5	1 (1.9)	>5, <10	0 (0.0)
	>0.5, <1	1 (1.9)	≥10	52 (100.0)
	<0.5	48 (92.3)		

## Data Availability

The data generated or analyzed during this study are included in this published article and its additional files. Some of the datasets are available from the corresponding author upon reasonable request.

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
