# Peer review of "Predictors for False-Negative Interferon-Gamma Release Assay Results in Hemodialysis Patients with Latent Tuberculosis Infection"

_diagnostics, 2022, doi:10.3390/diagnostics13010088_

Round 1
Reviewer 1 Report
1. The resolution of the figures is not so high. They are obscure although I can see the information. It would be better if the authors provide figures with higher resolution.
2. The annotated information of y-axis in Figure 1 and 2 is not correct. It should be “IFN-γ”, instead of “IFN-¥ã”. Please revise it.
3. In lines 159-161, the discussion section, the authors mentioned a limitation of this study the number and the age distribution in each study group do not match. May the age distribution difference affect the results of this study? How do the authors plan to address this limitation?
4. In line 61, in vivo and ex vivo should be in italics.
5. In table 1, What is “IHD”? Please provide the full name when it appears in the manuscript first.
6. The English of this manuscript is not so good. For example, the sentence should be “The number of samples and the age distribution in each study group do not match.”
Author Response
Author’s response: We deeply appreciate your time and careful review. In accordance with your comments, we have made the appropriate revisions to the manuscript
- The resolution of the figures is not so high. They are obscure although I can see the information. It would be better if the authors provide figures with higher resolution.
Author’s response: We deeply appreciate the reviewer’s kind and detailed comments. We added it by increasing the resolution of all Figures.
- The annotated information of y-axis in Figure 1 and 2 is not correct. It should be “IFN-γ”, instead of “IFN-¥ã”. Please revise it.
Author’s response: We appreciate the reviewer for the insightful comment. In the process of modifying the resolution of the figure, the annotation information on the y-axis was also modified.
(page 6, Figure 1, line 207, in revised manuscript)
(page 6, Figure 2, line 214, in revised manuscript)
- In lines 159-161, the discussion section, the authors mentioned a limitation of this study the number and the age distribution in each study group do not match. May the age distribution difference affect the results of this study? How do the authors plan to address this limitation?
Author’s response: The main point of this study is the discussion of the possibility of IGRA false negativity due to lower immunity in HD patients. Therefore, to minimize the deformation of differentiated immunity of individuals in each group, it is thought that more reliable research results will be produced if the study is conducted in the same population as the age group possible.
In a further study, ATB patients, HD patients, and healthy individuals will minimize variation by group by excluding abnormal individuals after pre-immune function tests such as NK cell activity or WBC diff count. In addition, it is expected that more reliable results will be obtained if the study is conducted with the same number of recruitment groups for each group.
- In line 61, in vivo and ex vivo should be in italics.
Author’s response: In accordance with the reviewer’s comments, we revised it to in vivo and ex vivo italics in the manuscript according to the reviewer's opinion. In addition, in the process of modification, the ex vivo of line 66 was also modified in italics.
- In table 1, What is “IHD”? Please provide the full name when it appears in the manuscript first.
Author’s response: We appreciate the reviewer for the insightful comment. Table 1's IHD is not an important factor in this study, so it was excluded from the manuscript.
- The English of this manuscript is not so good. For example, the sentence should be “The number of samples and the age distribution in each study group do not match.”
Author’s response: We appreciate the reviewer for the insightful comment. we revised the overall typographical errors in the manuscript.
(page 3, line 125, in revised manuscript)
(page 5, line 185-186, in revised manuscript)
(page 9, line 279, in revised manuscript)
(page 10, line 296, in revised manuscript)

Reviewer 2 Report
The title indicates predictors of slow immune response in HD patients with LTBI, while the authors describe and conclude that T cell responses are reduced as compared to healthy subjects which may lead to under-detection of LTBI in HD patients, with no mention of indicators for the slow response. The authors may consider a more appropriate title for the study
WHO references may be updated to recent ones, for example, the WHO global TB report 2022
Author Response
The title indicates predictors of slow immune response in HD patients with LTBI, while the authors describe and conclude that T cell responses are reduced as compared to healthy subjects which may lead to under-detection of LTBI in HD patients, with no mention of indicators for the slow response. The authors may consider a more appropriate title for the study
Author’s response: We deeply appreciate your time and careful review. In accordance with your comments, we have made the appropriate revisions to the manuscript. We fully agree with your comments. However, since this study discusses the possibility of IGRA false negatives due to abnormal T-cell reactions with a mitogen-nil value of 10 or less, we think this is a variable to be confirmed in IGRA tests.
WHO references may be updated to recent ones, for example, the WHO global TB report 2022:
Author’s response: In accordance with the reviewer’s comments, we updated the WHO global TB report 2022.
(page 10, line 320, in revised manuscript)
(page 10, line 323, in revised manuscript)

Round 2
Reviewer 1 Report
1. The format of Table 1 is inappropriate. Why are there two lines below?